# Risk factor associations for severe COVID-19, influenza and pneumonia in people with diabetes to inform future pandemic preparations: UK population-based cohort study

Rhian Hopkins ,[1] Katherine G Young ,[1] Nicholas J Thomas,[1] James Godwin,[1] Daniyal Raja,[1] Bilal A Mateen,[2,3] Robert J Challen,[4,5] Sebastian J Vollmer,[6] Beverley M Shields ,[1] Andrew P McGovern ,[1] John M Dennis [1]

Oral presentation of preliminary results from this study have been given at the Diabetes UK Professional Conference 2022 and the European Association for the Study of Diabetes (EASD) Annual Meeting 2022.

APM and JMD are joint senior authors.

For numbered affiliations see end of article.

**Correspondence to**
Rhian Hopkins;
rh530@exeter.ac.uk

## ABSTRACT

**Objective** This study aimed to compare clinical and sociodemographic risk factors for severe COVID-19, influenza and pneumonia, in people with diabetes.
**Design** Population-based cohort study.
**Setting** UK primary care records (Clinical Practice Research Datalink) linked to mortality and hospital records.
**Participants** Individuals with type 1 and type 2 diabetes (COVID-19 cohort: n=43 033 type 1 diabetes and n=584 854 type 2 diabetes, influenza and pneumonia cohort: n=42 488 type 1 diabetes and n=585 289 type 2 diabetes).
**Primary and secondary outcome measures** COVID-19 hospitalisation from 1 February 2020 to 31 October 2020 (pre-COVID-19 vaccination roll-out), and influenza and pneumonia hospitalisation from 1 September 2016 to 31 May 2019 (pre-COVID-19 pandemic). Secondary outcomes were COVID-19 and pneumonia mortality. Associations between clinical and sociodemographic risk factors and each outcome were assessed using multivariable Cox proportional hazards models. In people with type 2 diabetes, we explored modifying effects of glycated haemoglobin (HbA1c) and body mass index (BMI) by age, sex and ethnicity.
**Results** In type 2 diabetes, poor glycaemic control and severe obesity were consistently associated with increased risk of hospitalisation for COVID-19, influenza and pneumonia. The highest HbA1c and BMI-associated relative risks were observed in people aged under 70 years. Sociodemographic-associated risk differed markedly by respiratory infection, particularly for ethnicity. Compared with people of white ethnicity, black and south Asian groups had a greater risk of COVID-19 hospitalisation, but a lesser risk of pneumonia hospitalisation. Risk factor associations for type 1 diabetes and for type 2 diabetes mortality were broadly consistent with the primary analysis.
**Conclusions** Clinical risk factors of high HbA1c and severe obesity are consistently associated with severe outcomes from COVID-19, influenza and pneumonia, especially in younger people. In contrast, associations with sociodemographic risk factors differed by type

## STRENGTHS AND LIMITATIONS OF THIS STUDY

⇒ Uses a large primary care dataset of people with both type 1 and type 2 diabetes with linked hospital and mortality records.
⇒ Many potential risk factors, including potentially modifiable factors such as HbA1c and body mass index (BMI), assessed and included in models guided by previous studies and clinical knowledge.
⇒ Sociodemographic risk factor associations, in particular ethnicity differences, may be influenced by factors unique to the COVID-19 pandemic, and knowledge of these differences could inform future risk stratification.
⇒ Limitations include the potential for misclassification of infection outcomes; however, associations using different outcome definitions were robust in sensitivity analyses.
⇒ Associations do not have a causal interpretation but are still useful for identifying those at high risk.

of respiratory infection. This emphasises that risk stratification should be specific to individual respiratory infections.

## INTRODUCTION

People with diabetes are twice as likely to die from infection compared with the general population.[1] Infections are also a major cause of morbidity in people with diabetes, with one in four hospitalisations in men and one in three hospitalisations in women now being attributable to infections.[2] The burden of respiratory infections is increasing, comprising at least 10% of hospitalisations in people with diabetes in 2018, compared with less than 4% in 2003.[2] During the COVID-19 pandemic, diabetes was one of the most common comorbidities associated with severe infection and poor outcomes.[3–11] People with

type 2 diabetes were twice as likely to die with COVID-19 in hospital as the general population, with the risk being even higher in those with type 1 diabetes.[11] This is concordant with other severe respiratory infections including pneumonia, for which an excess risk of hospitalisation has been observed in people with type 2 diabetes,[12] and influenza, for which an increased risk of severe infection has been observed in older adults with diabetes.[13]

Despite the high burden of respiratory infection in people with diabetes, there has been no previous population-based comparison of risk factor associations across major respiratory infections. Previous studies have identified clear heterogeneity in the risk of severe COVID-19 in people with diabetes,[14–17] with a higher risk in those of older age, male sex, non-white ethnicity and higher social deprivation,[16 17] and differences by potentially modifiable risk factors, for example, increased risk with higher glycated haemoglobin (HbA1c) and higher body mass index (BMI).[14 16–19] Although broadly similar differences in the risk of other severe respiratory infections by sex,[20–24] age,[20–22] deprivation,[20 22 25] ethnicity,[26] glycaemic control[12 27] and BMI[28] have been suggested in studies of people with and without diabetes, the consistency of these risk factor associations across respiratory infections has not been robustly established. The limited studies to-date comparing COVID-19 risk factors to influenza and pneumonia have used selectively recruited cohorts, focused on the specific risk factors of sex and obesity,[29 30] or used positive COVID-19 tests/incident infection (influenced by country-specific testing policies) as the outcome.[31] As recently highlighted by the Chief Medical Officer for England, research to provide a robust understanding of the comparability of risk factors for major respiratory infections is urgently needed,[32] and could inform future risk stratification and targeted intervention.

In this study in people with diabetes, we aimed to compare clinical and sociodemographic risk factors for COVID-19 hospitalisation and mortality, with risk factors for influenza and pneumonia. We also aimed to assess heterogeneity in potentially modifiable HbA1c and BMI associated risk by age, sex and ethnicity.

## METHODS

### Data source

We used UK population-based data from the Clinical Practice Research Datalink (CPRD) Aurum, a large database of longitudinal, routinely collected medical records covering demographics, diagnoses, prescriptions and test results.[33] CPRD Aurum data are sourced from over 2000 primary care general practitioner (GP) practices from across the UK,[34] representing 13% of the population and is largely representative of the broader UK population.[33] We extracted data on all individuals with diabetes actively registered with a GP practice from 1 September 2016 to 31 October 2020. Primary care records were linked to national records of hospital admissions (Hospital Episode

Statistics (HES)) and deaths (Office for National Statistics). This combined dataset is one of the largest primary and secondary care datasets in the world and gave us a near-complete healthcare and mortality record for over half a million people with diabetes. We also included linked Index of Multiple Deprivation data (the official national measure of deprivation).

### Study population

#### Inclusion and exclusion criteria

We included people with type 1 diabetes and type 2 diabetes with valid linkage data (online supplemental figure 1). We classified type 1 diabetes and type 2 diabetes using an algorithm based on diabetes clinical code counts, insulin prescriptions, oral hypoglycaemic agent (OHA) prescriptions and diagnosis age (algorithm available at https://github.com/Exeter-Diabetes/CPRD-Codelists). The accuracy of this classification approach has recently been confirmed against the use of genetic and biochemical biomarkers of diabetes type.[35] Only those diagnosed with type 2 diabetes aged 20 or over were included to reduce the impact of potential misclassification in young adults. No age restrictions were applied for people with type 1 diabetes.

### Cohorts

To study COVID-19, we identified people with diabetes actively registered on 1 February 2020 (COVID-19 baseline date), who were followed up until 31 October 2020 (pre-COVID-19 vaccination roll-out, covering the first wave of COVID-19 and start of the second wave). To study influenza and pneumonia infection outcomes, we defined a pre-COVID-19 pandemic study period, including individuals with diabetes actively registered on 1 September 2016 (influenza/pneumonia baseline date) followed up until 31 May 2019. This study period enabled assessment of influenza and pneumonia outcomes over multiple recent UK influenza seasons prior to COVID-19.

### Outcomes

Primary outcomes were first hospitalisation for COVID-19, influenza or pneumonia during the respective study periods. Outcomes were identified using International Classification of Diseases 10th revision (ICD-10) codes for the infection of interest (online supplemental table 1) recorded as a diagnosis during hospital admission in HES. Secondary outcomes were cause-specific mortality, defined as an ICD-10 code for the infection of interest recorded as any cause of death, or hospitalisation for the infection of interest with the discharge method or location recorded as death.

### Risk factors and covariates

The risk factors of interest comprised baseline sociodemographic features (sex, age, ethnicity, deprivation), diabetes features (duration of diabetes, HbA1c, microvascular complications (diabetic nephropathy, neuropathy and retinopathy)) and BMI. We also included a wide range of covariates in our analysis: smoking status,

major comorbidities (cardiovascular, respiratory, neurological and oncological conditions, as well as chronic kidney disease, and recent hospitalisation (respiratory infection, any other cause)), diabetes treatments (no treatment, OHA prescription only, insulin), medications affecting the immune response (immunosuppressants, oral steroids), medications used for long-term respiratory conditions (leukotrienes, long-acting β agonists) and geographical region. Each comorbidity and medication were included as individual binary variables and missing data were treated as the absence of that covariate. All other covariates were included as categorical variables, and any missing values were grouped into a separate category. Full definitions of each variable are provided in online supplemental table 2, all were defined using primary care and, where appropriate, HES records (codes available at https://github.com/Exeter-Diabetes/CPRD-Codelists).

## Statistical analysis
### Primary analysis
Multivariable Cox proportional hazards models were used to identify risk factor associations for COVID-19, influenza and pneumonia hospitalisation, with proportional hazards assumptions for each variable assessed by visual inspection of Schoenfeld residuals. People were followed up until the earliest of: the outcome of interest, deregistration from GP, death or the end of each study period. Hazard ratios (HRs) for the potential risk factors in the model were calculated for each infection separately. For people with type 2 diabetes, we fitted separate multivariable models for each infection outcome including all potential risk factors (sex, age, ethnicity, deprivation, diabetes duration, HbA1c, number of microvascular complications, BMI) and covariates (smoking status, comorbidities, diabetes treatment, other medications, and region (full analysis set) (online supplemental table 2)). As the cohort of people with type 1 diabetes was much smaller, we used a reduced risk factor set comprising sex, age, ethnicity, deprivation, diabetes duration, HbA1c and BMI, with adjustment for region only (restricted analysis set). Analysis of mortality outcomes in people with type 1 diabetes, and mortality from influenza in people with type 2 diabetes, was not performed due to a limited number of deaths in these groups.

### HbA1c and BMI specific associations
In people with type 2 diabetes, we assessed the associations of continuous HbA1c and BMI with hospitalisation for each infection outcome, adjusted for the full analysis set. To allow for non-linearity, restricted cubic splines were fitted with 5 knots for COVID-19 and pneumonia and with 3 knots for influenza, with knot selection informed by Akaike information criterion. To explore heterogeneity in risk across sociodemographic subgroups, we refitted the same models including an interaction between continuous HbA1c/BMI and subgroups defined by age (over 70 years old/under 70 years old), sex (male/female), and ethnicity (white/black/south Asian ethnicity). We

grouped age into over and under 70 years in order to be consistent with previous research of COVID-19 risk factors in people with diabetes.[17] Statistical significance of subgroup interactions were tested using analysis of variance. In people with type 1 diabetes, the same approach as for type 2 diabetes was used to assess continuous associations of HbA1c and BMI with hospitalisation outcomes, adjusted for the restricted analysis set. Subgroup analysis was not performed due to low numbers.

### Sensitivity analysis
We repeated the primary analysis with restricted outcome definitions where the infection of interest was recorded as a primary diagnosis during the hospital admission or the primary cause of death. To assess the potential impact of influenza and pneumonia vaccination on our findings, we repeated the primary analysis in subgroups with and without a history of vaccination for influenza in the 2 years prior to baseline, and pneumococcal vaccination prior to baseline.

## Patient and public involvement
Patients and the public were not involved in this study.

## RESULTS
### Type 2 diabetes
A total of 584 854 people with type 2 diabetes were evaluated for COVID-19 outcomes, of whom 5965 (1.02%) were hospitalised with COVID-19 from 1 February 2020 to 31 October 2020. A total of 585 289 people with type 2 diabetes were evaluated for pneumonia and influenza outcomes, of whom 38 088 (6.51%) were hospitalised with pneumonia, and 3226 (0.55%) were hospitalised with influenza from 1 September 2016 to 31 May 2019. Online supplemental table 3 details the frequency of ICD-10 codes recorded for each infection hospitalisation outcome. The baseline characteristics of the two cohorts were similar (table 1 and online supplemental table 4). The majority of hospitalisations occurred in the older age groups for all three infections (table 1). The median age of those hospitalised was 74.8 years for COVID-19, 75.6 years for influenza and 79.4 years for pneumonia.

#### Poor glycaemic control and severe obesity are consistent risk factors for COVID-19, influenza and pneumonia in type 2 diabetes
High HbA1c>86 mmol/mol (>10%) was consistently associated with increased hospitalisation for COVID-19, influenza and pneumonia (figure 1). When assessed continuously, there was an increasing risk of COVID-19 and influenza hospitalisation above HbA1c 53 mmol/mol (7%), but a more gradual risk increase for pneumonia (figure 2). Compared with an HbA1c of 53 mmol/mol (7%), an HbA1c of 75 mmol/mol (9%) was associated with a 15% increased risk of COVID-19 (HR 1.15, 95% CI 1.06 to 1.26), 12% increased risk of influenza (HR 1.12, 95% CI 1.07 to 1.18) and 6% increased risk of pneumonia hospitalisation (HR 1.06, 95% CI 1.02 to 1.10).

**Table 1** Baseline clinical and sociodemographic characteristics in each cohort and COVID-19, influenza and pneumonia hospitalisations in type 2 diabetes

| | 2020 Cohort | COVID-19 hospitalisations | 2016 Cohort | Influenza hospitalisations | Pneumonia hospitalisations |
|---|---|---|---|---|---|
| Number of individuals | 584854 | 5965 | 585289 | 3226 | 38088 |
| Mean (SD) follow-up, days | | 265.0 (38.3) | | 915.2 (224.8) | 897.4 (246.3) |
| **Sex** | | | | | |
| Female | 253360 (43.3) | 2317 (38.8) | 256656 (43.9) | 1581 (49.0) | 17140 (45.0) |
| Male | 331494 (56.7) | 3648 (61.2) | 328633 (56.1) | 1645 (51.0) | 20948 (55.0) |
| **Age group, years** | | | | | |
| <40 | 12898 (2.2) | 70 (1.2) | 13468 (2.3) | 46 (1.4) | 164 (0.4) |
| 40–49 | 46075 (7.9) | 258 (4.3) | 49649 (8.5) | 128 (4.0) | 717 (1.9) |
| 50–59 | 114537 (19.6) | 729 (12.2) | 113285 (19.4) | 329 (10.2) | 2495 (6.6) |
| 60–69 | 151493 (25.9) | 1198 (20.1) | 153782 (26.3) | 679 (21.0) | 6068 (15.9) |
| 70–79 | 155424 (26.6) | 1660 (27.8) | 152401 (26.0) | 1078 (33.4) | 12073 (31.7) |
| 80–89 | 89871 (15.4) | 1671 (28.0) | 88492 (15.1) | 811 (25.1) | 13324 (35.0) |
| 90+ | 14556 (2.5) | 379 (6.4) | 14212 (2.4) | 155 (4.8) | 3247 (8.5) |
| **Ethnicity** | | | | | |
| White | 445160 (76.1) | 4200 (70.4) | 457714 (78.2) | 2586 (80.2) | 33206 (87.2) |
| South Asian | 77253 (13.2) | 891 (14.9) | 71050 (12.1) | 403 (12.5) | 2935 (7.7) |
| Black | 35144 (6.0) | 645 (10.8) | 33175 (5.7) | 156 (4.8) | 1324 (3.5) |
| Other | 9886 (1.7) | 144 (2.4) | 8178 (1.4) | 35 (1.1) | 306 (0.8) |
| Mixed | 6267 (1.1) | 69 (1.2) | 5654 (1.0) | 37 (1.1) | 246 (0.6) |
| Unknown | 11144 (1.9) | 16 (0.3) | 9518 (1.6) | 9 (0.3) | 71 (0.2) |
| **Index of multiple deprivation quintile** | | | | | |
| 1 (least deprived) | 102950 (17.6) | 761 (12.8) | 103184 (17.6) | 502 (15.6) | 6296 (16.5) |
| 2 | 106968 (18.3) | 921 (15.4) | 109006 (18.6) | 548 (17.0) | 6923 (18.2) |
| 3 | 113454 (19.4) | 1088 (18.2) | 113661 (19.4) | 601 (18.6) | 7264 (19.1) |
| 4 | 127048 (21.7) | 1446 (24.2) | 125492 (21.4) | 698 (21.6) | 8199 (21.5) |
| 5 (most deprived) | 134146 (22.9) | 1747 (29.3) | 133563 (22.8) | 873 (27.1) | 9380 (24.6) |
| Missing | 288 (0.0) | 2 (0.0) | 383 (0.1) | 4 (0.1) | 26 (0.1) |
| **Duration of diagnosed diabetes, years** | | | | | |
| <1 | 14311 (2.4) | 67 (1.1) | 36503 (6.2) | 146 (4.5) | 1401 (3.7) |
| 1–2 | 60219 (10.3) | 397 (6.7) | 77510 (13.2) | 306 (9.5) | 3324 (8.7) |
| 3–5 | 105094 (18.0) | 796 (13.3) | 111199 (19.0) | 491 (15.2) | 5218 (13.7) |
| 6–9 | 128817 (22.0) | 1116 (18.7) | 125083 (21.4) | 621 (19.2) | 7088 (18.6) |
| 10–14 | 124471 (21.3) | 1250 (21.0) | 126421 (21.6) | 771 (23.9) | 9620 (25.3) |
| 15–19 | 89704 (15.3) | 1225 (20.5) | 62373 (10.7) | 465 (14.4) | 5764 (15.1) |
| 20+ | 62238 (10.6) | 1114 (18.7) | 46200 (7.9) | 426 (13.2) | 5673 (14.9) |
| **HbA1c, mmol/mol** | | | | | |
| <48 | 167739 (28.7) | 1679 (28.1) | 175659 (30.0) | 934 (29.0) | 12271 (32.2) |
| 48–53 | 116015 (19.8) | 1054 (17.7) | 116131 (19.8) | 557 (17.3) | 7015 (18.4) |
| 53–64 | 147209 (25.2) | 1352 (22.7) | 142662 (24.4) | 791 (24.5) | 8912 (23.4) |
| 64–75 | 66071 (11.3) | 767 (12.9) | 62291 (10.6) | 412 (12.8) | 4062 (10.7) |
| 75–86 | 33970 (5.8) | 408 (6.8) | 32257 (5.5) | 184 (5.7) | 2191 (5.8) |
| 86+ | 39727 (6.8) | 580 (9.7) | 37470 (6.4) | 270 (8.4) | 2646 (6.9) |
| Missing | 14123 (2.4) | 125 (2.1) | 18819 (3.2) | 78 (2.4) | 991 (2.6) |

Continued

**Table 1** Continued

| | 2020 Cohort | COVID-19 hospitalisations | 2016 Cohort | Influenza hospitalisations | Pneumonia hospitalisations |
|---|---|---|---|---|---|
| Number of microvascular complications | | | | | |
| 0 | 264011 (45.1) | 1889 (31.7) | 281811 (48.1) | 1077 (33.4) | 11930 (31.3) |
| 1 | 216115 (37.0) | 2140 (35.9) | 204550 (34.9) | 1256 (38.9) | 14444 (37.9) |
| 2 | 96948 (16.6) | 1621 (27.2) | 91671 (15.7) | 789 (24.5) | 10288 (27.0) |
| 3 | 7780 (1.3) | 315 (5.3) | 7257 (1.2) | 104 (3.2) | 1426 (3.7) |
| BMI, kg/m$^2$ | | | | | |
| <18.5 | 3026 (0.5) | 66 (1.1) | 2801 (0.5) | 16 (0.5) | 497 (1.3) |
| 18.5–24.9 | 86199 (14.7) | 1009 (16.9) | 79018 (13.5) | 476 (14.8) | 6985 (18.3) |
| 25–29.9 | 184626 (31.6) | 1747 (29.3) | 178501 (30.5) | 885 (27.4) | 10950 (28.7) |
| 30–34.9 | 151732 (25.9) | 1350 (22.6) | 151161 (25.8) | 809 (25.1) | 8249 (21.7) |
| 35–39.9 | 74766 (12.8) | 752 (12.6) | 75339 (12.9) | 405 (12.6) | 3976 (10.4) |
| 40+ | 48455 (8.3) | 517 (8.7) | 48685 (8.3) | 302 (9.4) | 2877 (7.6) |
| Missing | 36050 (6.2) | 524 (8.8) | 49784 (8.5) | 333 (10.3) | 4554 (12.0) |

BMI, body mass index.

There was a U-shaped association with BMI for COVID-19 and pneumonia, and obesity was associated with an increased risk of hospitalisation for all three respiratory infections (figure 1). There was a stronger association between increasing BMI and COVID-19 hospitalisation than hospitalisation for influenza or pneumonia (figure 3). Compared with a BMI of 30 kg/m$^2$, a BMI of 35 kg/m$^2$ was associated with an 19% increased risk of COVID-19 (HR 1.19, 95% CI 1.10 to 1.27), 8% increased risk of influenza (HR 1.08, 95% CI 1.05 to 1.11) and a 6% increased risk of pneumonia hospitalisation (HR 1.06, 95% CI 1.03 to 1.09).

An increasing number of microvascular complications was also associated with increased hospitalisation for all three respiratory infections (figure 1). Diabetes duration was not associated with hospitalisation for all three infections, except in those with short duration (< 1 year) where there was reduced risk for COVID-19 and pneumonia (figure 2). Online supplemental figure 2 and table 5 report the full set of risk factors studied.

### HbA1c and BMI associated risk differs by age and sex

In subgroup analysis, we observed a difference in HbA1c associations by age but not sex; there was a stronger association between higher HbA1c and hospitalisation in those aged under 70 compared with those over 70 for COVID-19 and pneumonia, but not influenza (figure 2). There was little evidence for a difference by sex (figure 2). For BMI, there was a stronger association with increasing BMI and hospitalisation in those aged under 70 and in women for all three infections (figure 3). We observed no clear difference in HbA1c or BMI associations by ethnicity, although numbers were limited (online supplemental figure 3A,B).

### Non-white ethnicity, male sex and increased deprivation are more important risk factors for COVID-19 hospitalisation than pneumonia or influenza hospitalisation in type 2 diabetes

Across the three respiratory infections, we observed differential effects of ethnicity, male sex and higher deprivation on hospitalisation risk. Compared with people of white ethnicity, black and south Asian groups had a higher risk of COVID-19 hospitalisation, but a lower risk of pneumonia hospitalisation (figure 1). For influenza, a higher risk was seen in the south Asian group only. Male sex was more strongly associated with increased COVID-19 hospitalisation than pneumonia hospitalisation, and was associated with decreased hospitalisation for influenza (figure 1). For all three infections, higher deprivation was associated with a higher risk of hospitalisation, but there was a stronger association with COVID-19 than influenza or pneumonia (figure 1). Older age was strongly associated with pneumonia hospitalisation, whereas a lesser age-associated gradient was seen for COVID-19 and influenza (figure 1).

### Risk factors associations are similar for mortality

Of a total of 584854, 2635 (0.45%) died from COVID-19, and 12652 of 585289 (2.16%) died from pneumonia in the respective study periods (baseline characteristics: online supplemental table 6). Clinical and sociodemographic risk factor associations for COVID-19 and pneumonia mortality were largely consistent with those for hospitalisation (online supplemental figure 4).

### Type 1 diabetes

Of people with type 1 diabetes, 209 of 43033 (0.49%) were hospitalised with COVID-19 during the 2020 study period. Of 42488, 208 (0.49%) were hospitalised for influenza, and 1401 of 42488 (3.30%) were hospitalised

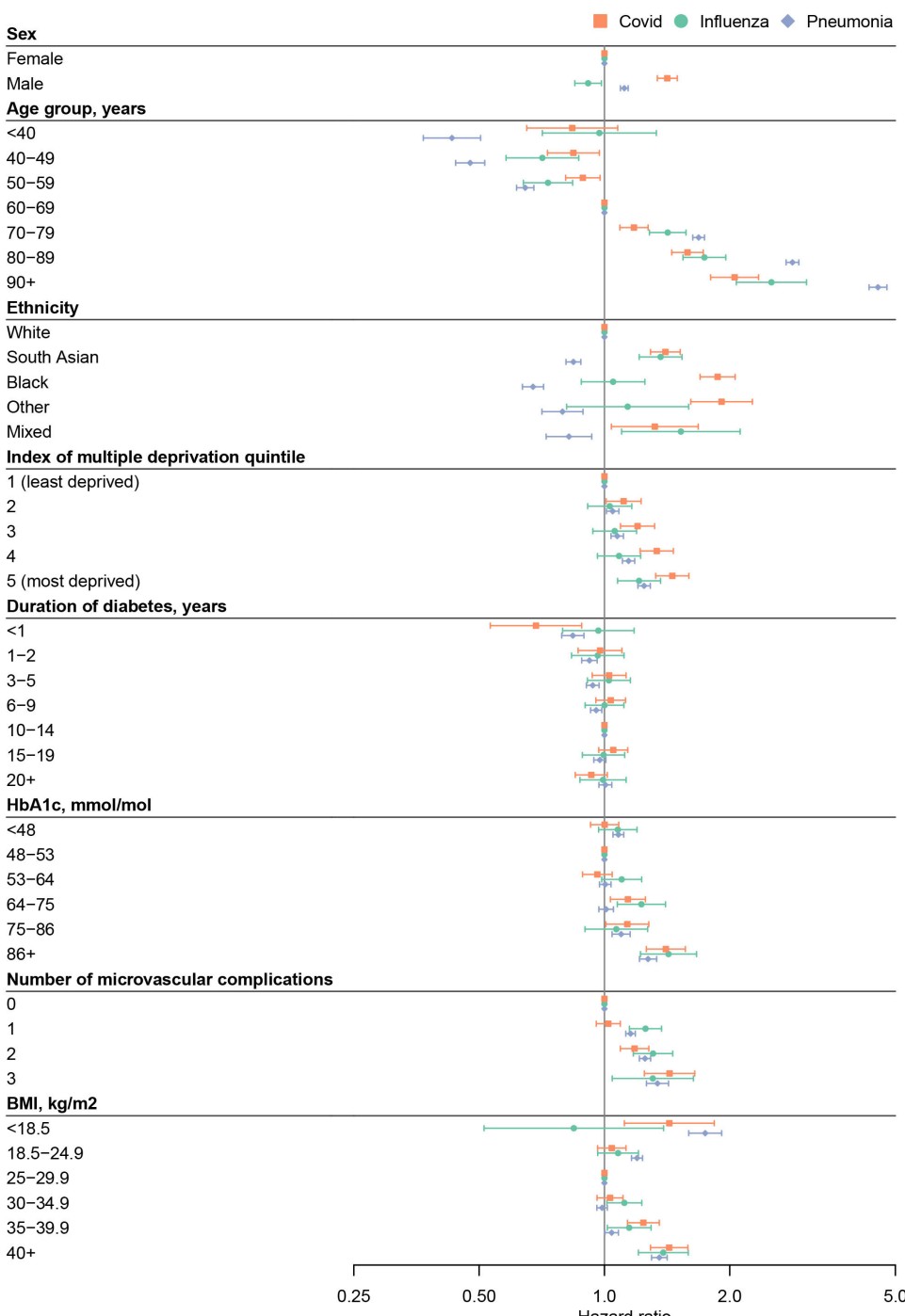

**Figure 1** HRs and 95% CIs for the association of clinical and sociodemographic risk factors with hospitalisation for COVID-19 (orange squares), influenza (green circles) and pneumonia (blue diamonds) in type 2 diabetes

with pneumonia during the 2016–2019 study period (baseline characteristics: online supplemental table 7). Similarly to type 2 diabetes, higher HbA1c and higher BMI were associated with increased hospitalisation for COVID-19, influenza and pneumonia in type 1 diabetes (online supplemental figures 5 and 6A,B). Sociodemographic associations were broadly similar to those in type 2 diabetes, although the association between increasing age and pneumonia hospitalisation was not stronger than for COVID-19, and there was no clear difference in COVID-19 risk by deprivation quintile in type 1 diabetes.

We observed an increased risk of COVID-19 hospitalisation in people of non-white ethnicities, and in people of south Asian ethnicity for influenza, but no difference in risk by ethnicity for pneumonia (online supplemental figure 5).

## Sensitivity analyses
Results were largely consistent with the primary analyses when outcomes were restricted to primary hospitalisation diagnoses (type 1 diabetes: online supplemental table 8 and figure 7; type 2 diabetes: online supplemental table

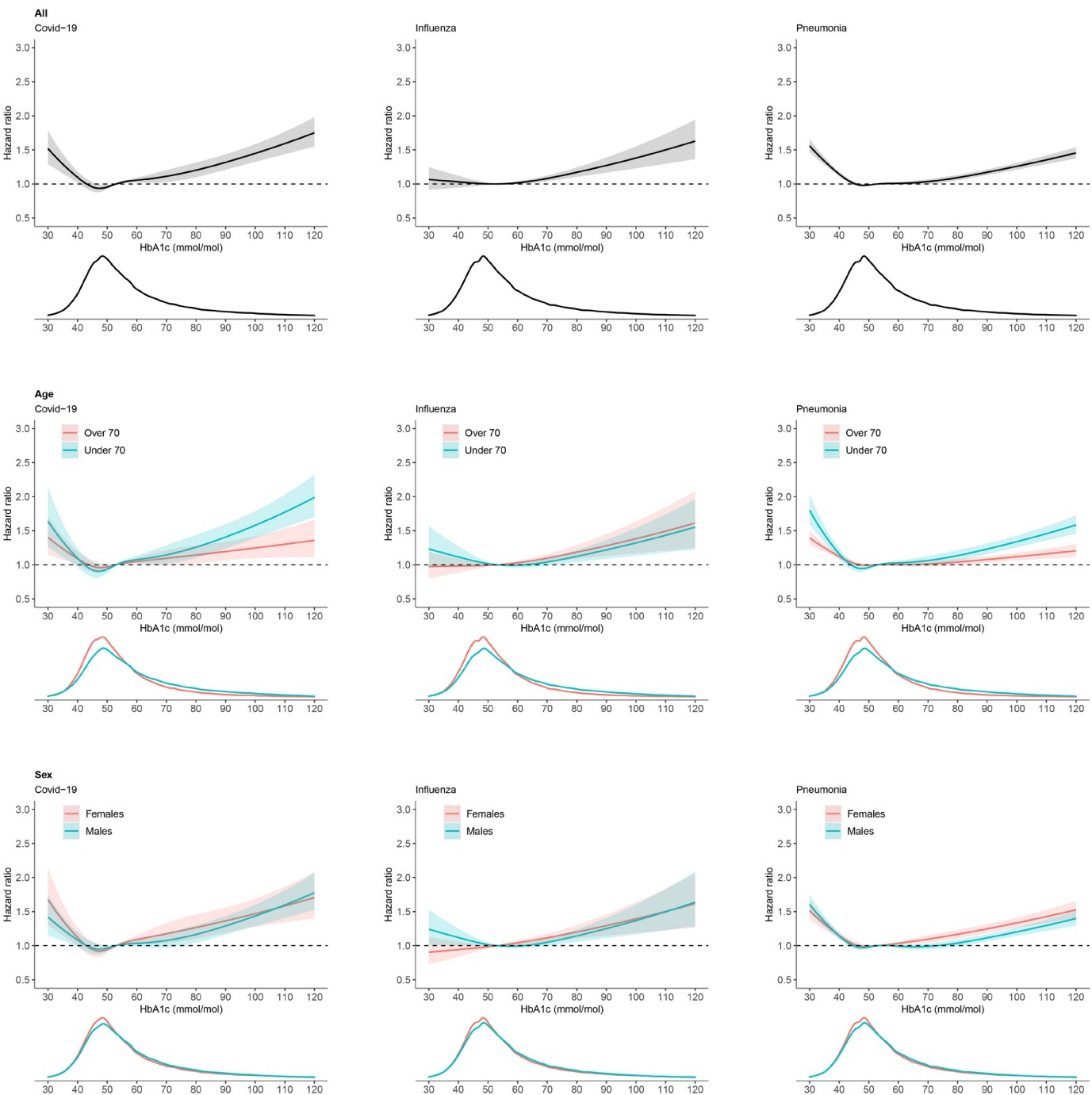

**Figure 2** HRs and 95% CIs for the association of continuous HbA1c with hospitalisation for COVID-19, influenza and pneumonia in type 2 diabetes. Overall, and by age and sex. Statistical significance of age subgroup interactions with HbA1c: p=0.0190 for COVID-19, p=0.2214 for influenza, and p<0.0001 for pneumonia. Statistical significance of sex subgroup interactions with HbA1c: p=0.5488 for COVID-19, p=0.0737 for influenza and p=0.0037 for pneumonia. Density plots show the distribution of HbA1c in each group.

9 and figure 8), and primary cause of death in type 2 diabetes (online supplemental table 10 and figure 9). When repeating the primary analysis in those vaccinated and unvaccinated, associations with pneumonia hospitalisation were consistent in those with and without a pneumococcal vaccination (online supplemental figure 10). There were very low numbers of people without an influenza vaccination; however, associations with influenza hospitalisation appeared to be consistent in those with

and without vaccination for influenza (online supplemental figure 11).

**DISCUSSION**

In a large-scale UK population-based analysis, we provide a comprehensive comparison of risk factors for COVID-19, influenza and pneumonia in people with type 1 and type 2 diabetes. We show that although clinical risk factor

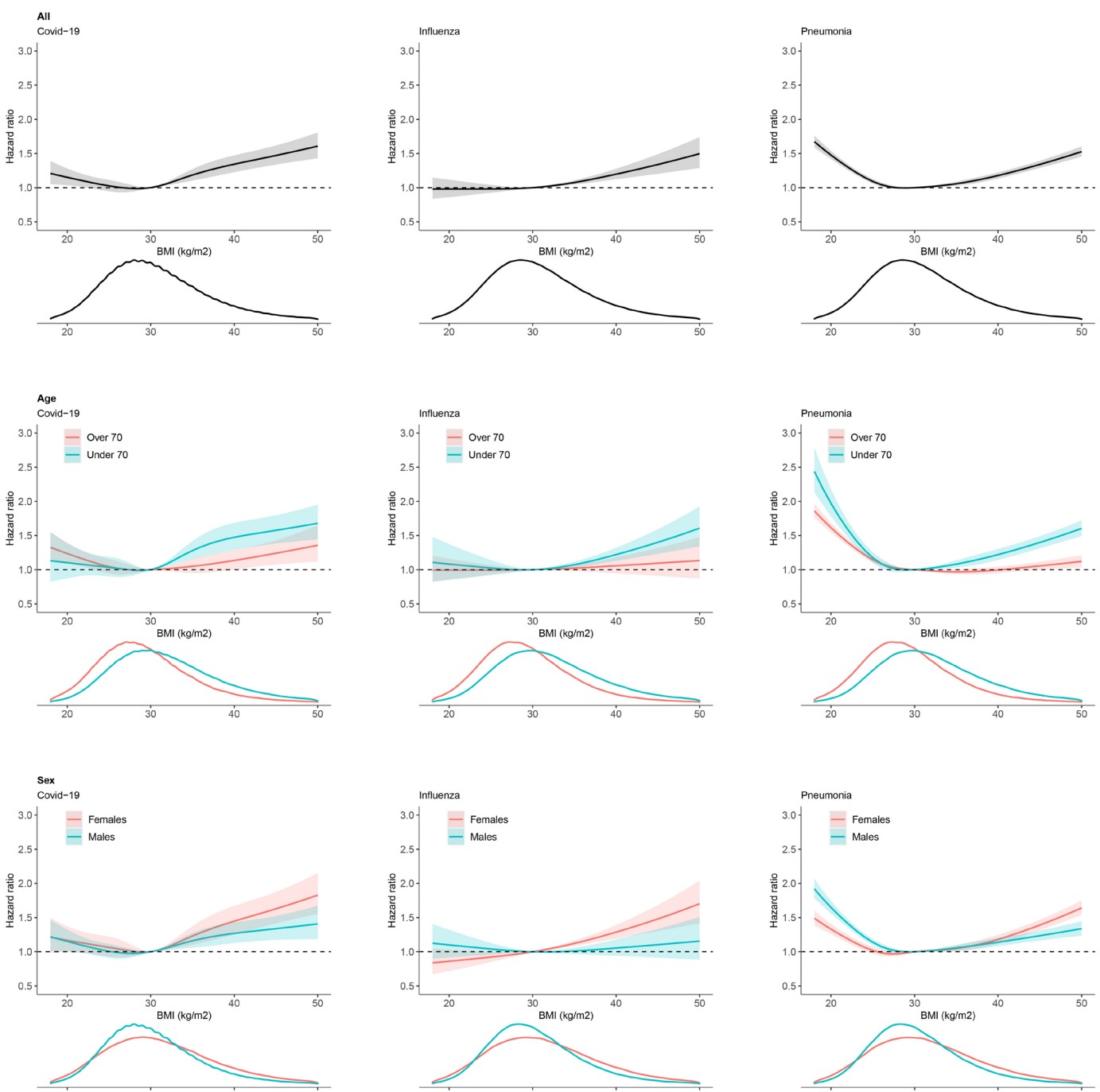

**Figure 3** HRs and 95% CIs for the association of continuous body mass index (BMI) with hospitalisation for COVID-19, influenza and pneumonia in type 2 diabetes. Overall, and by age and sex. Statistical significance of age subgroup interactions with BMI: p=0.0046 for COVID-19, p=0.0934 for influenza, and p<0.0001 for pneumonia. Statistical significance of sex subgroup interactions with BMI: p=0.2780 for COVID-19, p=0.0008 for influenza and p=<0.0001 for pneumonia. Density plots show the distribution of BMI in each group.

associations for HbA1c, BMI and microvascular complications are similar for all three respiratory infections, there is evidence that risk varies meaningfully by age and sex. Notably, while poor glycaemic control and marked obesity are consistently associated with an increased risk of hospitalisation and death across all three infections, in relative terms these risk increases are greater in those aged under 70. For BMI, the association between higher

BMI (>30) and a severe outcome is consistently higher for women than men for all 3 infections.

We also demonstrate clear differences in sociodemographic risk factor associations by respiratory infection type. The previously reported higher risk of severe COVID-19 in people of non-white ethnicity[10 17 36] is not consistent with other respiratory infections and is the opposite pattern of association seen for pneumonia.

Additionally, male sex and higher deprivation appear to be more important risk factors for COVID-19 than influenza and pneumonia. Findings have important implications for the management of modifiable infection risk factors in people with diabetes as well as current and future resource planning to manage severe respiratory infection.

While COVID-19 risk factors in people with diabetes have previously been robustly described,[16 17] previous population-based studies have not compared risk factor associations with those seen for other major respiratory infections. To our knowledge, only one general population (with and without diabetes) study has compared a range of risk factors for COVID-19 to influenza and pneumonia. However, this study used a selected cohort (UK Biobank) and a positive test result/incident infection as the outcome measure which means findings are likely to be influenced by UK testing policies. Due to the under-representation of non-white ethnicities in the UK Biobank, this study also had limited power to describe associations by ethnicity.[31]

The associations between higher HbA1c and BMI and severe COVID-19 outcomes identified in this study are consistent with previous studies of people with diabetes.[14 16–19] While UK population-based data have suggested that associations between poor glycaemic control and obesity and COVID-19 mortality were greater in people younger than 70,[17] our study extends this previous work by evaluating non-linear continuous associations for HbA1c and BMI for severe COVID-19, and showing associations are not specific to COVID-19. For pneumonia, poor glycaemic control has been associated with increased hospitalisation[12] but risk heterogeneity by patient phenotype has not been previously evaluated. Our finding of a markedly increased risk of severe COVID-19, influenza and pneumonia in women with higher BMI is supported by prospective cohort data on the whole UK Biobank population, including individuals with and without diabetes.[29]

Notably, we show that previously described risk factor associations between male sex, older age, non-white ethnicity, increasing deprivation and severe COVID-19 outcomes in people with diabetes[16 17] are not generalisable to influenza and pneumonia. Previous studies describing associations between sex and pneumonia have not reported consistent effects,[37] and evidence is also lacking for risk factors associated with severe influenza with the exception of increasing age.[38] Previous studies have found evidence of increased risk of severe COVID-19 outcomes in those of non-white ethnicity[10 17 36] consistent with our findings. While these studies have mainly been based on UK data, ethnic disparities in rates of influenza-associated hospitalisation have been observed in US data, with certain ethnic groups having poorer outcomes compared with those of white ethnicity.[26]

## Strengths and limitations

Strengths of our study include the large size of the dataset, representativeness to the UK population, and linkage to longitudinal hospital admission and mortality data allowing us to robustly define infection outcomes and systematically compare COVID-19, influenza and pneumonia risk factor associations. The large dataset allowed us to study both type 1 diabetes and type 2 diabetes. We evaluated associations for a range of risk factors and across different subgroups; however, low numbers in certain groups may limit the precision of those estimates. As with all studies of this nature, misclassification or misdiagnosis of our outcomes is possible and relies on correct coding in the records. Evaluating both hospitalisation and mortality outcomes, and analysis to evaluate the sensitivity of results to our outcome definitions, mitigated against this limitation. Despite this, we were unable to completely differentiate between infection being the reason for admission or death or being contracted during admission, or only contributing to hospitalisation/death. We have also noted some differences in the coding of the studied respiratory infections in the hospital records; COVID-19 and influenza were mostly confirmed by a positive test whereas pneumonia coding was much broader and less specific, which should be considered when interpreting the results. The associations we have found do not have a causal interpretation and we cannot rule out potential confounding by unmeasured variables, which is an inherent limitation of many observational studies; however, we have adjusted for a wide range of covariates. There is also potential for misclassification of diabetes type in our cohorts, but we have used comprehensive algorithms considering prescriptions, age at diagnosis as well as diabetes codes, to define diabetes type as thoroughly as possible. This approach has been validated against a biological diabetes definition using classification biomarkers.[35]

Due to our focus on severe infection outcomes, we will have missed milder community cases of infection; to assess risk factors for being infected would have required a different study design. The use of two different study periods for COVID-19 and influenza/pneumonia, while unavoidable, has a risk of bias and may have influenced the observed associations. The associations found may reflect differences in behaviour, in particular during the first wave of the COVID-19 pandemic, and be influenced by factors unique to this period including marked geographical differences in infection rates, as well interventions such as lockdowns and shielding. The disproportionate impact of these measures on non-white and more deprived groups[39] may have influenced the sociodemographic disparities we have observed. The risks associated with the disruption to routine care in people with diabetes during the COVID-19 pandemic[40] also make comparisons with a pre-pandemic study period difficult. Despite these limitations, a key strength of our analysis is the use of standardised case and risk factor definitions, and consistent statistical methodology, ensuring associations observed

across the two study periods were as comparable as possible. Using this approach, we demonstrate that major clinical risk factors are very similar across infections, a finding that is unlikely to be explained by specific impacts of the COVID-19 pandemic.

## Clinical implications

This study highlights the potential importance of good glycaemic control and supporting weight loss for managing risk of three major respiratory infections in people with diabetes. While BMI may be hard to modify at an individual level, a population-level investment in effective weight reduction measures may help reduce severe respiratory infection. The substantial differences in sociodemographic risk factor associations across the different respiratory infections suggest that we cannot assume high-risk groups for one respiratory infection are the same as another. This means that existing risk models for respiratory infection, such as those developed for COVID-19,[5 41] may not be applicable to new respiratory infections and so in the event of a new respiratory pathogen, early analysis of risk factors would be needed to correctly target interventions. The finding that the increased risk of severe COVID-19 in those of non-white ethnicity is largely not seen for other respiratory infections (except for an increase in severe influenza risk in people of south Asian ethnicity) highlights the importance of further exploration as to why ethnicity-associated infection risk appears to vary by infection type.

## Conclusion

Poor glycaemic control and obesity are associated with a consistently increased risk of severe outcomes from COVID-19, influenza and pneumonia, especially in those aged under 70. In contrast, even in a single country setting, sociodemographic risk factor associations differ by respiratory infection. This highlights the need for the development of population-specific risk stratification approaches for individual respiratory infections to ensure accurate identification of people with diabetes at the highest risk of severe outcomes.

**Author affiliations**
[1]Institute of Biomedical & Clinical Science, University of Exeter Medical School, Exeter, UK
[2]The Alan Turing Institute, London, UK
[3]Institute of Health Informatics, University College London, London, UK
[4]Engineering Mathematics, University of Bristol, Bristol, UK
[5]NIHR Applied Research Collaboration South West Peninsula, Exeter, UK
[6]Department of Statistics, University of Warwick, Coventry, UK

**Acknowledgements** This article is based on data from the Clinical Practice Research Datalink obtained under licence from the UK Medicines and Healthcare products Regulatory Agency. CPRD data are provided by patients and collected by the NHS as part of their care and support. Approval for CPRD data access and the study protocol was granted by the CPRD Independent Scientific Advisory Committee (Project-ID: 20_000101). This study was supported by the National Institute for Health and Care Research Exeter Biomedical Research Centre. The views expressed are those of the author(s) and not necessarily those of the NIHR or the Department of Health and Social Care or Diabetes UK. RH and KGY are supported by Research

England's Expanding Excellence in England (E3) fund. NJT is funded by a Wellcome Trust-funded GW4 PhD fellowship. RJC is funded through the AvonCAP study which is an investigator-led University of Bristol study funded by Pfizer and further supported by the National Institute for Health and Care Research Applied Research Collaboration South West Peninsula. BMS is supported by the NIHR Exeter Clinical Research Facility. JMD is supported by an Independent Fellowship funded by Research England's Expanding Excellence in England (E3) fund. The funders had no role in the study design, data collection, data analysis, data interpretation, writing of the article, or the decision to submit for publication.

**Contributors** The study concept and design were conceived and developed by RH, JMD, APM, BAM, SJV and NJT. JMD, KGY, RH and BMS had access to all the raw data sets used for the study. RH, KGY, DR and RJC prepared the data for analysis. RH undertook the analysis, with support from JMD, BMS, JG and APM. All authors provided support for the interpretation of results, critically revised the manuscript, and saw and approved the final article. JMD and APM attest that all listed authors meet authorship criteria, and that no others meeting the criteria have been omitted. JMD and APM were responsible for the decision to submit for publication, and are the guarantors of this work and, as such, had full access to all the data in the study and takes responsibility for the integrity of the data and the accuracy of the data analysis.

**Funding** This work was supported by Diabetes UK (20/0006220).

**Competing interests** BAM is an employee of Wellcome Trust; holds an honorary post at the Alan Turing Institute and University College London; and declares payments from the Medical Research Council, Health Data Research UK, British Heart Foundation, and Engineering and Physical Sciences Research Council (grant EP/N510129/). RJC declares research funding from Pfizer. SJV declares support from the University of Warwick, University of Kaiserslautern, and German Research Center for Artificial Intelligence; consulting fees from PUMAS; stock from Freshflow; and grant funding from The Alan Turing Institute (EP/N510129), Engineering and Physical Sciences Research Council, and Massachusetts Institute of Technology. APM received research funding from Eli Lilly and Company, Pfizer, and AstraZeneca. For all authors these are outside the submitted work; there are no other relationships or activities that could appear to have influenced the submitted work.

**Patient and public involvement** Patients and/or the public were not involved in the design, or conduct, or reporting, or dissemination plans of this research.

**Patient consent for publication** Not applicable.

**Ethics approval** The study protocol was approved by the CPRD Independent Scientific Advisory Committee (Project-ID: 20_000101).

**Provenance and peer review** Not commissioned; externally peer reviewed.

**Data availability statement** No data are available. No additional data are available from the authors although CPRD data are available by application to CPRD Independent Scientific Advisory Committee.

**ORCID iDs**
Rhian Hopkins http://orcid.org/0000-0001-6054-3582
Katherine G Young http://orcid.org/0000-0003-2570-3864
Beverley M Shields http://orcid.org/0000-0003-3785-327X
Andrew P McGovern http://orcid.org/0000-0002-6833-9399
John M Dennis http://orcid.org/0000-0002-7171-732X

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
