## [Reviewer comments · BMJ Open]

ARTICLE DETAILS

TITLE (PROVISIONAL)	Risk factor associations for severe Covid-19, influenza, and pneumonia in people with diabetes to inform future pandemic preparations: UK population-based cohort study
AUTHORS	Hopkins, Rhian; Young, Katherine; Thomas, Nick; Godwin, James; Raja, Daniyal; Mateen, Bilal; Challen, Robert; Vollmer, Sebastian; Shields, Beverley; McGovern, Andrew; Dennis, John

VERSION 1 – REVIEW

REVIEWER	Collier, Andrew Glasgow Caledonian University
REVIEW RETURNED	05-Sep-2023

GENERAL COMMENTS	Although a large cohort once divided up with different groups the numbers do fall significantly particularly with non SE Asian patients. Stats therefore cannot be as strong. Type 1 patients will be younger (much smaller group) and show different associations. My concern is - what is really new and how will that inform future management? - possibly weight loss and better glycaemic control (which we knew). The authors say that 70 years appears to be the age that associations change - why? This needs discussed. The results will be skewed because young unwell patients may well stay at home and therefore will not be counted in the analysis. This needs acknowledged.
---

REVIEWER	Athanasopoulos, Stavros National and Kapodistrian University of Athens School of Medicine, Clinical Therapeutics
REVIEW RETURNED	17-Sep-2023

GENERAL COMMENTS	Very well written manuscript addressing important question not only during the pandemic but also thereafter for future infections. The figures illustrate well your findings and your discussion is to the point.
---

VERSION 1 – AUTHOR RESPONSE

Reviewer: 1
Dr. Andrew Collier, Glasgow Caledonian University
Comments to the Author:

Although a large cohort once divided up with different groups the numbers do fall significantly particularly with non SE Asian patients. Stats therefore cannot be as strong.

>>Author response: Thank you for taking the time to consider our paper. We acknowledge that low numbers in certain subgroups is a limitation of this study, although it is worth noting that this is one of the largest multi-ethnic combined primary and secondary care datasets available for research globally. However, we agree this is an important limitation and therefore have attempted to address it by adding the following text to the strengths and limitations section of our discussion:

“We evaluated associations for a range of risk factors and across different subgroups, however low numbers in certain groups may limit the precision of those estimates.”

Type 1 patients will be younger (much smaller group) and show different associations.

>>As above we acknowledge that low numbers in the type 1 diabetes cohort is an important limitation of this study. Therefore, we did not replicate the subgroup analyses performed for type 2 diabetes and evaluated a reduced risk factor set in this cohort. However, we were still sufficiently powered to identify key clinical and sociodemographic risk factor associations in this group.

My concern is - what is really new and how will that inform future management? - possibly weight loss and better glycaemic control (which we knew).

>>This study is the first systematic comparison of risk factors for Covid-19 and other respiratory infections in people with diabetes. Our clinical implications section highlights the need for infection specific risk stratification approaches due to differences in high risk groups between infections, which will be important in the event of a new respiratory pathogen. We also highlight the importance maintaining good glycaemic control and supporting weight loss for managing risk of major respiratory infections.

The authors say that 70 years appears to be the age that associations change - why? This needs discussed.

>>We chose to use under 70 and over 70 years as age subgroups in order to be consistent with previous studies of Covid-19 risk factors in people with diabetes (Holman et al, 2017) and similarly showed a greater association between poor glycaemic control and severe Covid-19 outcomes in people younger than 70. We have added the following text to the methods section to highlight this point:

“We grouped age into over and under 70 years in order to be consistent with previous research of Covid-19 risk factors in people with diabetes [17].”

The results will be skewed because young unwell patients may well stay at home and therefore will not be counted in the analysis. This needs acknowledged.

>>We acknowledge this as a limitation of our study and highlight in the discussion that associations may reflect differences in behaviour (particularly for Covid-19) with the following text:

“ The associations found may reflect differences in behaviours, in particular during the first wave of the Covid-19 pandemic, and be influenced by factors unique to this period including marked geographical differences in infection rates, as well interventions such as lockdowns and shielding.”

Reviewer: 2

Dr. Stavros Athanasopoulos , National and Kapodistrian University of Athens School of Medicine
Comments to the Author:

Very well written manuscript addressing important question not only during the pandemic but also thereafter for future infections.

The figures illustrate well your findings and your discussion is to the point.

>>Author response: Thank you for taking the time to review our paper and for acknowledging the importance of this study for future respiratory infections.

VERSION 2 – REVIEW

REVIEWER	Collier, Andrew Glasgow Caledonian University
REVIEW RETURNED	22-Dec-2023
GENERAL COMMENTS	I am happy that the authors have answered all the concerns and questions posed.